# Memory impairment in chronic experimental Chagas disease: Benznidazole therapy reversed cognitive deficit in association with reduction of parasite load and oxidative stress in the nervous tissue

Glaucia Vilar-Pereira[1]☉, Leda Castaño Barrios[1]☉, Andrea Alice da Silva[2], Angelica Martins Batista[1], Isabela Resende Pereira[1], Otacílio Cruz Moreira[3], Constança Britto[3], Hílton Antônio Mata dos Santos[4,5], Joseli Lannes-Vieira[1]*

1 Laboratório de Biologia das Interações, Instituto Oswaldo Cruz/Fiocruz, Rio de Janeiro, RJ, Brazil, 2 Laboratório Multiusuário de Apoio à Pesquisa em Nefrologia e Ciências Médicas, Departamento de Patologia, Faculdade de Medicina, Universidade Federal Fluminense, Niterói, RJ, Brazil, 3 Laboratório de Biologia Molecular e Doenças Endêmicas, IOC/Fiocruz, Rio de Janeiro, RJ, Brazil, 4 Faculdade de Farmácia, Universidade Federal do Rio de Janeiro, Rio de Janeiro, RJ, Brazil, 5 Laboratório de Análise e Desenvolvimento de Inibidores Enzimáticos e Laboratório Multiusuário de Análises por RMN, Universidade Federal do Rio de Janeiro, Rio de Janeiro, RJ, Brazil

☉ These authors contributed equally to this work.
* lannes@ioc.fiocruz.br, joselilannes@gmail.com

## Abstract

Memory impairment has been associated with chronic Chagas disease (CD), a neglected tropical disease caused by the protozoan parasite *Trypanosoma cruzi*. In degenerative diseases, memory loss has been associated with increased oxidative stress, revealed as enhanced lipid peroxidation, in the cerebral cortex. Benznidazole (Bz), a trypanocidal drug efficient to reduce blood parasite load in the acute and chronic phases of infection, showed controversial effects on heart disease progression, the main clinical manifestation of CD. Here, we evaluated whether C57BL/6 mice infected with the Colombian type I *T. cruzi* strain present memory deficit assessed by (i) the novel object recognition task, (ii) the open field test and (iii) the aversive shock evoked test, at 120 days post infection (dpi). Next, we tested the effects of Bz therapy (25mg/Kg/day, for 30 consecutive days) on memory evocation, and tried to establish a relation between memory loss, parasite load and oxidative stress in the central nervous system (CNS). At 120 dpi, *T. cruzi*-infected mice showed memory impairment, compared with age-matched non-infected controls. Bz therapy (from 120 to 150 dpi) hampered the progression of habituation and aversive memory loss and, moreover, reversed memory impairment in object recognition. In vehicle-administered infected mice, neuroinflammation was absent albeit rare perivascular mononuclear cells were found in meninges and choroid plexus. Bz therapy abrogated the infiltration of the CNS by inflammatory cells, and reduced parasite load in hippocampus and cerebral cortex. At 120 and 150 dpi, lipid peroxidation was increased in the hippocampus and cortex tissue extracts. Notably, Bz therapy reduced levels of lipid peroxidation in the cerebral cortex. Therefore, in

**Data Availability Statement:** All relevant data are within the manuscript and its Supporting Information files.

**Funding:** This work was supported by grants from Fundação Carlos Chagas Filho de Amparo à Pesquisa do Estado do Rio de Janeiro/FAPERJ (E-26/110.153/2013, E-26/202.572/2019, E-26/210.190/2018) and the Brazilian Research Council/CNPq (BPP-304474/2015-0, PDJ- 159084/2014-8, BPP 306037/2019-0, INCTV, National Institute for Science and Technology for Vaccines), Grant PAEF2-IOC/Fiocruz. J. Lannes-Vieira, C. Britto and O. Cruz Moreira are research fellows of the Brazilian Research Council/CNPq and recognized with the grant Scientist (JLV, CB) and Young Scientist (OCM) of the State of Rio de Janeiro/FAPERJ. This study was financed in part by the "Coordenação de Aperfeiçoamento de Pessoal de Nível Superior do Brasil" (CAPES) - Finance Code 001.

**Competing interests:** The authors have declared that no competing interests exist.

experimental chronic *T. cruzi* infection Bz therapy improved memory loss, in association with reduction of parasite load and oxidative stress in the CNS, providing a new perspective to improve the quality of life of Chagas disease patients.

# 1. Introduction

Chagas disease (CD), a neglected tropical disease caused by the intracellular protozoan parasite *Trypanosoma cruzi*, afflicts 7–8 million people worldwide, most of them born and resident in Latin America [1]. The 2nd Consensus on Chagas disease estimates that 1.9 to 4.6 million individuals are infected with *T. cruzi* in Brazil. Ten to thirty years after infection, 20 to 30% of the patients progress to the cardiac form and 5 to 10% to the digestive form of CD [2]. Although a nervous form has been proposed by Carlos Chagas [3], the involvement of the central nervous system (CNS) in CD remains a matter of debate, mainly due to the lack of histopathological evidence [4, 5]. The non-recognition of a nervous form of CD hampers the adoption of more specific approaches, therapeutic strategies, and proposal of consensus to treat CD patients with behavioral alterations. Several studies described CNS impairment and behavioral alterations in chronic CD patients, attributed to left ventricular heart dysfunction, resulting in brain ischemia by hypoperfusion and/or embolic events [6]. A higher frequency of cognitive changes has been observed in patients with Chagas' heart disease, when compared to patients with other cardiac diseases [7]. *Post-mortem* analysis of the CNS of CD patients showed cerebral and cerebellar atrophy without neuroinflammation [8]. Later, a computed tomography-based study revealed that the cerebral atrophy was independent of a structural cardiac disease, raising the idea that brain atrophy may represent the main anatomical substrate of cognitive impairment in CD patients [9]. An association between *T. cruzi* infection and cognitive abnormalities has been detected using the mini-mental state examination score to test elderly CD patients, which were not mediated by CD-related electrocardiographic alterations or digoxin medication [10]. Furthermore, cognitive dysfunction in chronic CD patients has been supported by deficiency in orientation, attention, non-verbal reasoning, information processing and learning [11, 12]. More import, memory loss has been described in chronic chagasic infection [13–15].

Acute and chronically *T. cruzi*-infected rats present memory deficit [16]. Previously, we have described that in chronically *T. cruzi*-infected C57BL/6 mice depressive-like behavior, anxiety and motor coordination disorder occur independently of sickness behavior and neuromuscular disorder [17, 18]. The present study was carried out to evaluate the presence of memory impairment in chronically *T. cruzi*-infected mice, using the (i) memory habituation test in the open field, (ii) novel object recognition memory task and (iii) aversive shock evoked test. Further, we challenged the contribution of parasite load in the CNS in behavioral alterations, treating chronically infected mice with the trypanocidal drug benznidazole (Bz). Lastly, we investigated the impact of chronic *T. cruzi* infection and Bz therapy on the grade of CNS tissue lipid peroxidation, a biomarker of oxidative stress associated with memory dysfunction in a neurodegenerative disorder [19, 20].

# 2. Materials and methods

## 2.1 Ethics statement

The experimental procedures were carried out in strict accordance with the recommendations of the Guide for the Care and Use of Laboratory Animals of the Brazilian National Council of

Animal Experimentation (https://www.mctic.gov.br/mctic/opencms/institucional/concea/) and the federal law 11.794 (8 October 2008). The Ethical Commission on Animal Use of Fiocruz and Oswaldo Cruz Institute/Fiocruz (licenses LW10/14 and L006/2018) approved all the procedures used in this study. All presented data were obtained from three independent experiments (Experiment Register Book #53 and Book #73, LBI/IOC-Fiocruz).

## 2.2 Experimental groups, infection by *Trypanosoma cruzi*, clinical follow-up and obtaining of brain tissue

Experimental check list is described in Author´s Check List (S1 Checklist). Details of experimental protocol are described in Fig 1. Female of 5-7-week-old mice (total of 92 mice) of the C57BL/6 strain (H-2$^b$) were supplied by the Institute of Science and Technology in Biomodels (ICTB) of the Oswaldo Cruz Foundation (Fiocruz) and housed in the Experimental Animal Facility (CEA-CF unit/IOC). Immediately after arrival, mice were housed in polypropylene cages with Pinus sawdust, randomly grouped in 3–5 mice per cage (experiment 1) or 5 mice per cage (experiments 2 and 3). The cages were maintained in microisolators, and mice received grain-based chaw food and water *ad libitum*. To minimize stress, mice were kept in adaptation for 10–14 days in a plastic igloo-enriched cage, in conditions free of specific pathogens, with light and noise control. After adaptation, mice were infected, treated, and analyzed according to the experimental protocols (Fig 1). Groups were composed as follow: **Exp 1** – time point pre-therapy–analysis at 120 dpi (total 12 mice): non-infected controls (5 mice); *T. cruzi*-infected (7 mice); **Exp 2** (total 40 mice): Part 1– time point pre-therapy (analysis at 120 dpi; total 15 mice)—non-infected controls (5 mice); *T. cruzi*-infected (10 mice); Part 2 –time point post-therapy (analysis at 150 dpi; total 25 mice)—non-infected controls (5 mice); vehicle-treated *T. cruzi*-infected (10 mice); Bz-treated *T. cruzi*-infected (10 mice); **Exp 3** (total 40 mice): repetition of Exp 2. These groups were formed at mice arrival when cages were numbered and randomly sorted for the experimental infection, analysis at 120 dpi or for

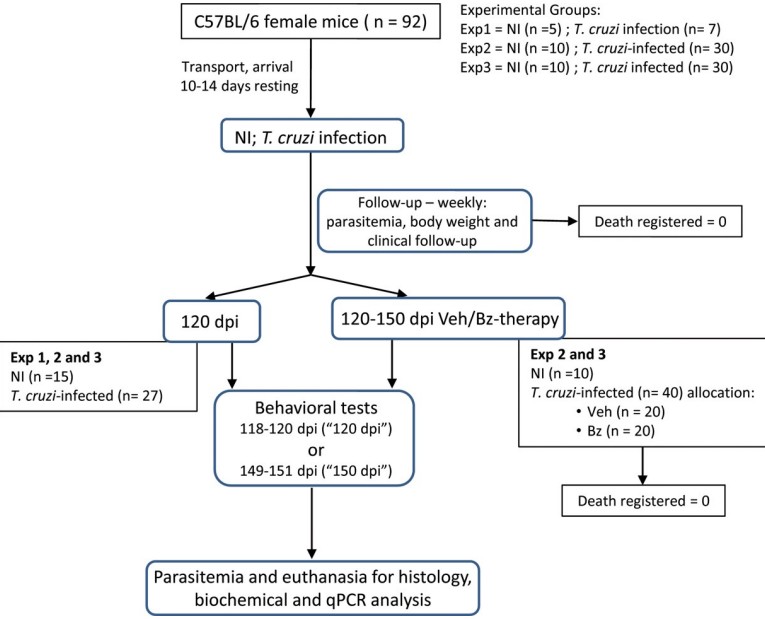

**Fig 1. Flow chart showing the experimental protocol with the number of animals used, death registered, and mice included in 3 (endpoint registered as "120 dpi") or 2 (endpoint registered as "150 dpi") independent experiments.**

treatments. The experimental procedures were performed by different investigators (cited here as initials): housing, cage numbering and experimental group sorting: JLV and IRP; infection: JLV; drug treatments: IRP; mice follow-up and behavior tests: GVP and LCB; Tissue obtaining: GVP, LCB, IRP, JLV; Tissue analysis: AAS, AMB, OCM, HAMS; Data analysis: GVP, LCB, AAS, AMB, IRP, OCM, CB, HAMS, JLV.

Mice were infected intraperitoneally with 100 trypomastigote forms of the Colombian strain suspended in 0.2 mL of sterile saline buffer. The trypomastigote forms used were isolated from a chronic CD patient, currently classified as Type I *T. cruzi* strain [21] and kept by serial passages in C57BL/6 female mice each 35 days, at the Laboratory of Biology of the Interactions (LBI/IOC). Parasitemia was performed weekly, as previously described [22]. Weekly, death was recorded, and clinical signs analyzed. For clinical follow-up we evaluated external physical conditions (piloerection, apathy, prostration, mobility, posture, aggressive behavior, pain), and body weight loss (which reveals loss of appetite) was assessed using a rodent weighing scale (Sartorius scale, ED623S-OCE, USA). Negative parasitemia (35 to 45 dpi) and any death outside of planned euthanasia or humane endpoints were the exclusion criteria. The humane endpoints criteria were body weight loss ($\geq$ 30% of the initial weight), injuries from fights, pain, posture, ataxia and immobility. According to the experimental protocols, mice were euthanized at the endpoints (120 and 150 dpi), using $CO_2$ inhalation in an appropriate chamber allowing 70% of $CO_2$ saturation for 2–3 minutes, followed by decapitation. The encephala were collected and weighted. Cortex and hippocampus were dissected [23], RNA later (AM7021, Thermo Fisher Scientific, USA) was added, and the tissues immediately stored at -80C for neurochemical assays: lipid peroxidation by evaluation of tissue levels of thiobarbituric acid reactive species and parasite load determination by quantitative PCR.

## 2.3 Benznidazole treatment

Groups of 7–10 infected animals were treated daily, from 120 to 150 dpi, by gavage with 0.1 mL of vehicle (Veh) apyrogenic vaccine-graded water (BioManguinhos, Fiocruz, Brazil) or 0.1 mL of Veh containing 25 mg/Kg of the trypanocidal medicament Bz (LAFEPE, Brazil), previously shown to be effective in controlling parasitemia and parasitism [24]. As controls, 3–10 sex- and age-matched non-infected (NI) mice were kept concurrently and submitted to Veh administration.

## 2.4 Conditions of behavioral evaluation

All behavioral tests were conducted between 8:00 am-3:00 pm and recorded using a DSC-DVD810 video camera (Sony, USA). To minimize stress and increase familiarity, all behavioral tests applied to different experimental groups were performed in an environment provided with 12 hours of light and 12 hours of dark cycle at a room temperature of 22 ± 2°C and noise level of approximately 40 dB produced by an air conditioner. Different groups of animals were submitted to behavioral tests from 118–120 dpi (referred as 120 dpi) or 149–151 dpi (referred as 150 dpi). No animals were re-tested. When possible, the animals were reused in different tests, aiming to reduce the number of animals used. Behavioral tests were performed from less stressful tests to the more stressful (habituation memory test, novel object recognition memory test, passive aversive shock evoked test, grip strength meter test).

## 2.5 Habituation memory test

To evaluate the habituation memory, we used the open field apparatus, a white acrylic arena measuring 60 cm x 60 cm. The floor of the apparatus was divided by black grid lines into 49 squares of approximately 8.5 cm each and two imaginary areas—the periphery (40 squares

along the walls) and center (9 squares in the central area of the apparatus). In the training session, the animals were carefully placed in the rear left square of the device and left to explore the environment freely for 5 min. Immediately after this time, the animals were returned to the housing cages. The long-term memory test was performed 24-hours after the training, in which the procedure was repeated, and the 5 min session was recorded using a digital video camera (Sony, USA). The apparatus was cleaned with 70% alcohol and dried with gauze between tests. The memory retention was evaluated by counting the number of total lines crossed on the test session [25] and the individual baseline differences were corrected using the change ratio score to compare behavior during initial and final sessions, as follows: number of crossed lines day 2 /(number of crossed lines day 1 + number of crossed lines day 2), as previously described [26].

## 2.6 Novel object recognition memory task

The one-trial learning or one-trial object recognition task consisted of a sample trial of 5 min duration, in which mice explore two equal objects in a 60 cm x 60 cm open field arena (the same used for the habituation memory test). After a 24-hours intersession interval, a novel object is presented together with one familiar object already explored during the sample trial, for a 5 min duration test session. The apparatus and objects were cleaned with a solution of 70% alcohol and dried to eliminate the odor and trace of the previously tested mouse. This test is based on the innate preference of non-infected normal rodents to explore the novel object rather than the familiar one [27]. This kind of test is derived from the "visual paired-comparison paradigm" used in human and non-human primates and may allow interspecies comparison [28]. The discrimination index (DI) was calculated as follows: time exploring the novel object/(time exploring the novel object + time exploring the familiar object), as previously shown [29].

## 2.7 Aversive shock evoked test

The procedure was modified from the previously described test [30]. Briefly, the inhibitory avoidance apparatus (EP 104MR, Insight, Brazil) consisted of a 35 x 28 x 50 cm epoxy-painted 2 mm aluminum box with acrylic front door, whose floor consisted of parallel stainless-steel bars (3 mm diam.) spaced 1 cm apart. A 7 cm wide x 2.5 cm high platform was placed on the floor of the box against the wall of the righthand side. In a pre-exposure session, animals were placed on the platform and allowed to explore the box freely for 5 min without foot shock. A training session was carried out 2 hours after pre-exposure, followed by a test session 24-hours after training. For training, animals were placed on the platform and their latency to step down on the grid with all four paws was measured with an automatic device. In training sessions, immediately after stepping down on the grid, the animals received a 3.0-sec scrambled foot shock (0.6 mA), and one to five stimulations were required for memory acquisition. In test sessions, no foot shock was administered, and the step-down latency (maximum 120 sec) was used as a measure of memory retention. An increase in step-down latency during test was taken as an index of improved memory and vice versa.

## 2.8 Grip strength meter test

Muscle strength was assessed using the grip strength meter (EFF 305, Insight, Brazil), according to manufacturer's instructions. This noninvasive method, which may disclose a neuromuscular disorder, is based on the natural tendency of mice to grab a horizontal metal bar when slightly pulled by the tail for 2–3 seconds. The bar is attached to a force transducer which

measures the traction peak (in gram-force) that is displayed on a digital screen. Data are presented as mean of strength intensity = gram-force (gf)/body weight (g).

## 2.9 Histopathology

At 150 dpi, non-infected and *T. cruzi*-infected mice were euthanized, as described in item 2.2. The encephalon was removed, fixed in buffered formalin 10%, dehydrated and embedded in paraffin. Five μm-thick sections were prepared, stained with hematoxylin and eosin and two sections per encephalon tissue were blindly examined using light microscopy. Representative images were digitized using a Sight DS-U3 color-view digital camera adapted to an Eclipse Ci-S microscope and analyzed with the digital morphometric apparatus NIS Elements BR version 4.3 software (Nikon Co., Japan).

## 2.10 Determination of parasite load in the CNS by quantitative PCR (qPCR)

Mice were euthanized and encephalon removed and dissected, as described in item 2.2. DNA was extracted from dissected cerebral cortex and hippocampus samples using 1 mL of TRI-Reagent (Sigma-Aldrich, St. Louis, MO, USA) for each tissue (20 mg), 200 μL of chloroform (Merck, USA) were added and incubated for 2 min. After centrifugation at 12,000 x g at 4˚C for 15 min, the organic phase and interface were obtained, and the DNA was precipitated by adding 300 μL of 100% ethanol (Merck, USA) and incubated for 3 min. The material was centrifuged at 2000 x g for 5 min at 4˚C. The supernatant was discarded, and the DNA pellet was washed twice in 1 mL of a solution containing 0.1 M sodium citrate (Sigma-Aldrich, St. Louis, MO, USA) in 10% ethanol: incubation for 30 min at room temperature with periodic stirring and centrifuged at 2000 x g for 5 min at 4˚C. The DNA pellet was suspended in 1.5 mL of 75% ethanol and incubated for 15 min at room temperature with periodic shaking. After centrifugation at 2000 x g for 5 min at 4˚C, the supernatant was removed, and the DNA was dried for 10 min. Next, the DNA was dissolved in 100 μL of 8 mM NaOH (Sigma-Aldrich, St. Louis, MO, USA). Centrifuged at 13000 x g for 10 min and the supernatant aliquoted. The qPCR assays were performed by absolute quantification to estimate the parasite load, based on a standard curve produced from DNA samples extracted from fragments of mouse cortex or hippocampus artificially contaminated with *T. cruzi*. For this, 20 mg of cortex or hippocampus samples (uninfected mouse) were spiked with $10^5$ *T. cruzi* trypomastigotes, and DNA was extracted as described. Serial dilutions of 1:10 were carried out using Tris-EDTA buffer to produce the standard curve, ranging from $10^5$ to 1 parasite equivalents, and from 20 to $2 \times 10^{-4}$ mg mouse cortex or hippocampus equivalents. The qPCR assays were carried out with 5 μL of DNA, using the FastStart Universal Probe Master Mix (Roche Diagnostics, Mannheim, Germany) in a final volume of 20 μL. The amplifications were carried out in a ABI 7500 Fast Real Time PCR system (Applied Biosystems, USA) using 750 nM of Cruzi 1 (Sequence: 5 'AST CGG CTG ATC GTT TTC GA 3') and Cruzi 2 (Sequence: 5 'AAT TCC TCC AAG CAG CGG ATA 3') primers, 50 nM of Cruzi 3 probe (Sequence: 5'-FAM CAC ACA CTG GAC ACC AA- NFQ-MGB-3'). Concurrently, a TaqMan assay targeting mouse GAPDH (Glyceraldehyde-3-phosphate dehydrogenase VIC/TAMRA-labelled, Mm99999915g1, Thermo Fisher Scientific, USA) was used to quantify the brain tissue mg equivalents, according to manufacturer´s instructions. Thus, the parasite load was normalized by the tissue mg equivalents, by dividing the *T. cruzi* quantity by the mass of mice tissue equivalents (cortex or hippocampus). PCR cycling conditions were:

95˚C for 10 min, followed by 40 cycles at 95˚C 15s and 58˚C for 1min. To analyze the results, the threshold was set at 0.02.

## 2.11 Lipid peroxidation evaluation in the CNS

Mice were euthanized and encephalon removed and dissected, as described in item 2.2. The tissue levels of thiobarbituric acid reactive species (TBARS) in the cortex and hippocampus were determined by Ohkawa's method [31]. Malondialdehyde (MDA) is one of several low-molecular-weight end products of degradation of hydroperoxides and lipid peroxides formed during the oxidation of fatty acids [19, 32]. MDA reacts with thiobarbituric acid (TBA) to form a pink-colored dimeric compound. Dissected cortex and hippocampus were homogenated. Ten percent (w/v) of tissue homogenate was mixed with 8.1% sodium dodecyl sulfate (SDS), 20% acetic acid pH 3.5 and 0.8% TBA, and incubated at 95˚C for 1 hour. After incubation, the reaction product was extracted with n-butanol (1:1) and read using a spectrophotometer (Spectra Max M5, Molecular Devices, USA) at a wavelength of 532 nm.

## 2.12 Statistical analysis

The sample size was determined based on the experience of our group and previous studies using the model of experimental chronic chagasic cardiomyopathy; therefore, no formal sample size was calculated. The data of 3 independent experiments were grouped. Data are expressed as arithmetic means ± standard error (SE). Statistical comparisons between groups were carried out by analysis of variance (ANOVA) followed by Bonferroni *post-hoc* test. All statistical tests were performed with GraphPad Prism 8.0 (La Jolla, CA, USA). Differences were considered statistically significant when $p < 0.05$.

## 3. Results

### 3.1 Muscular strength is preserved in C57BL/6 mice chronically infected with the Colombian *Trypanosoma cruzi* strain

Out of 92 C57BL/6 female mice, 25 were non-infected (NI) controls and 67 were infected with 100 trypomastigote forms of the Colombian strain of *T. cruzi* and weekly monitored for parasitemia, body weight and clinical follow-up (Fig 1). In the first set of experiments, mice were analyzed at 120 dpi (Fig 2A) when 100% of them were alive (Fig 2B), and no mouse was excluded by death outside of planned euthanasia or humane endpoint criteria. At 120 dpi, NI controls and infected mice presented similar physical characteristics and body weight (S1A Fig; 22.7 ± 0.70 in NI mice *vs* 20.6 ± 2.9 in *T. cruzi*-infected mice; $p > 0.05$). After 15 dpi, all infected mice showed circulating parasites, blood parasitism peaked at 42–45 dpi, and at 120 dpi parasitemia was controlled (Fig 2C). In comparison with sex- and age-matched NI controls, *T. cruzi*-infected mice showed no muscular strength alteration ($p > 0.05$; Fig 2D). Therefore, chronically Colombian-infected C57BL/6 mice were able to perform behavioral tests based on mobility and body activity.

### 3.2 Memory dysfunctions and brain atrophy are detected in chronically *Trypanosoma cruzi*-infected mice

Habituation memory was assessed using the open field test and comparing the performance in the first and second day of testing. At 120 dpi, *T. cruzi*-infected mice and NI controls showed similar discrimination index ($p > 0.05$; Fig 3A), supporting that habituation memory was preserved. However, the performance of *T. cruzi*-infected mice in the novel

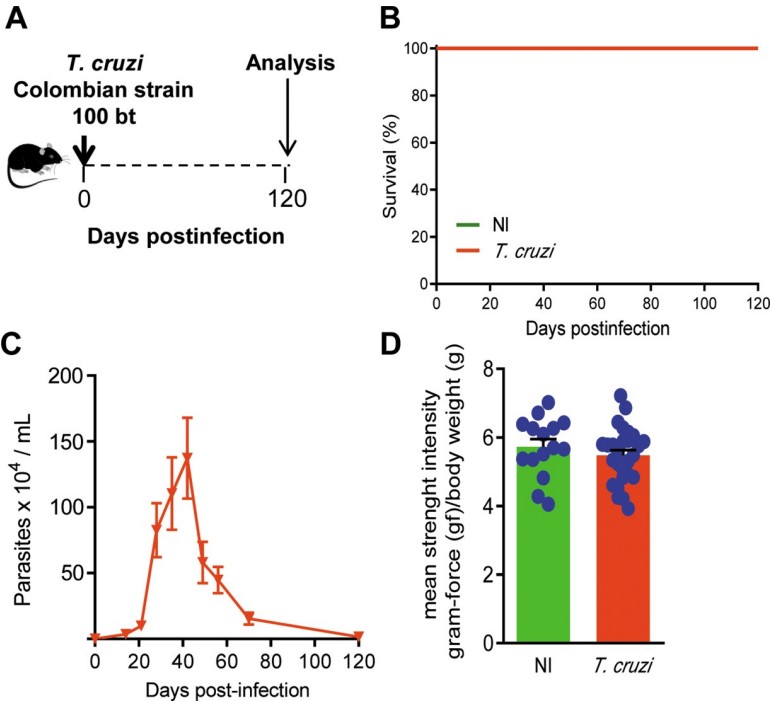

**Fig 2. C57BL/6 mice infected with the Colombian *Trypanosoma cruzi* strain control parasitemia, survive, and develop chronic phase with preserved muscle strength.** (**A**) Mice were infected with 100 blood trypomastigote forms, parasitemia and death were recorded weekly, and the animals were analyzed in the chronic phase of infection (120 dpi). (**B**) Survival curve (percentage of alive mice). (**C**) Parasitemia curve (Parasites x $10^4$/mL). (**D**) The graph shows the results of muscle strength [gram force (gf)/body weight (g)] of *T. cruzi*-infected mice compared with sex- and age-matched noninfected controls (NI), at 120 dpi. Each experimental group consisted of 5 NI mice and 7–10 *T. cruzi*-infected mice. Each circle represents an individual mouse. Data are represented as means ± SE of three independent experiments (15 NI; 27 *T. cruzi*-infected mice). Data were analyzed using *t*-Student test.

object recognition test revealed reduced discrimination index, supporting loss of object recognition memory (Fig 3B). At 120 dpi, similar numbers of stimulation were required for aversive memory acquisition in the training day (2.5 ± 0.34 in NI mice *vs* 1.9 ± 0.48 in *T. cruzi*-infected mice; $p > 0.05$). Interestingly, the aversive memory was preserved in most of the *T. cruzi*-infected mice (81%), though in 19% of them latency was reduced in the aversive shock evoked test (Fig 3C). At 120 dpi, no difference was detected in body weight of *T. cruzi*-infected mice, compared with NI controls (S1A Fig). However, reduced brain weight was observed in chronically *T. cruzi*-infected mice, in comparison with age-matched NI controls (Fig 3D), suggesting the presence of encephalon atrophy in chronic experimental chagasic infection.

### 3.3 Benznidazole therapy beneficially impacts on memory loss in chronically *Trypanosoma cruzi*-infected C57BL/6 mice

Chronically infected mice were treated with 25 mg/Kg/day of Bz from 120 to 150 dpi (Fig 4A) to evaluate the impact of Bz therapy on memory loss onset and progression. Survival was of 100% in the groups of Veh- and Bz-treated mice, and no mouse showed sickness scores for humane endpoint. At 150 dpi, Bz therapy did not impact body weight (S1A Fig), when compared to NI controls and Veh-treated infected and likened to *T. cruzi*-infected mice pre-therapy (at 120 dpi). At 120 and 150 dpi, *T. cruzi*-infected mice

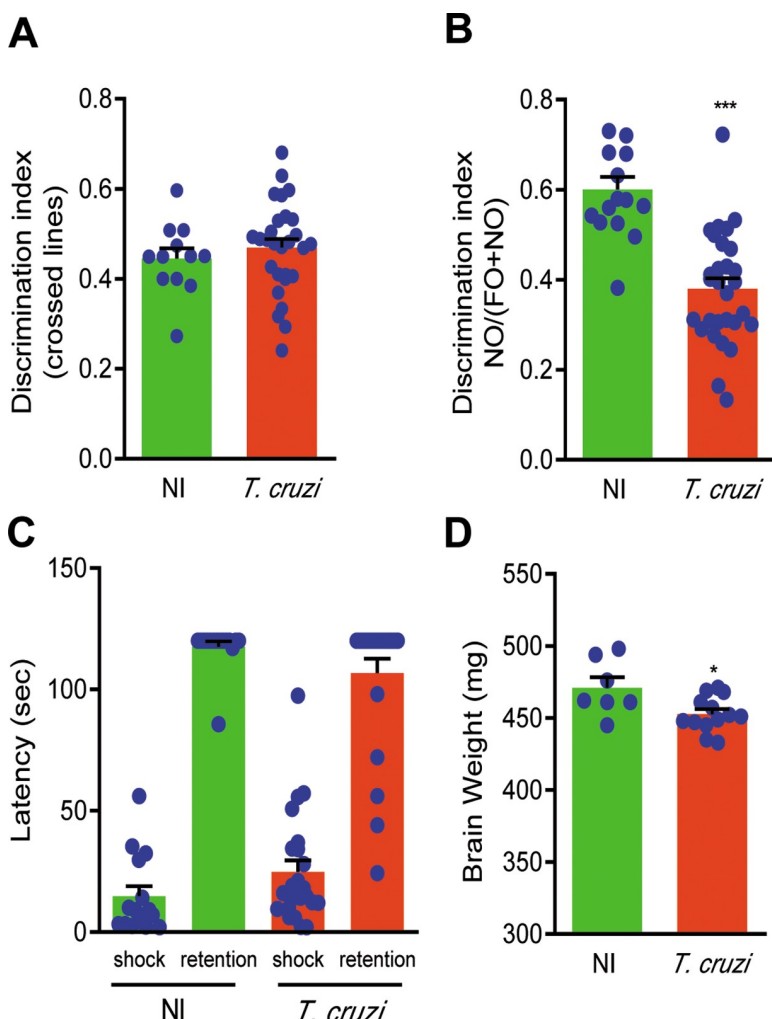

**Fig 3. Memory loss in chronically *Trypanosoma cruzi*-infected C57BL/6 mice.** Mice were infected with 100 blood trypomastigote forms of the Colombian *T. cruzi* strain and analyzed at 120 dpi, comparing with noninfected controls (NI). (**A**) The graph shows the discrimination index [number of crossed line day 2 /(number of crossed line day 1 + number of crossed line day 2)] assessed by the open field test used to evaluate habituation memory. (**B**) The graph shows the discrimination index [time exploring the novel object/(time exploring the novel object + time exploring the familiar object)] evaluated by the novel object recognition test object applied to assess recognition memory. (**C**) The graph shows the latency (second; sec) measured using the aversive shock evoked test to evaluate aversive memory. (**D**) The graph shows the encephalon weight (mg). Each experimental group consisted of 5 NI mice and 7–10 *T. cruzi*-infected mice. Each circle represents an individual mouse. Data are represented as means ± SE of three independent experiments (15 NI; 27 *T. cruzi*-infected mice). Data were analyzed using *t*-Student test (**A**, **B** and **D**) and ANOVA-Bonferroni posttest (**C**). *, $p < 0.05$, ***, $p < 0.001$, comparing *T. cruzi*-infected and NI mice.

presented increased relative spleen weight, and Bz administration reduced this spleno-megaly (S1B Fig), confirming the effectiveness of this therapy [24]. At 150 dpi, in comparison with sex- and age-matched NI controls, memory loss was detected in Veh-treated *T. cruzi*-infected mice submitted to the open field test (Fig 4B), novel object recognition task (Fig 4C) and shock evoked test (Fig 4D). Bz therapy initiated at 120 dpi prevented the habituation memory impairment (Fig 4B). Importantly, Bz treatment reversed the memory loss assessed by novel object recognition memory task (Fig 4C). The aversive memory loss already detected in a low number of mice (19%) at 120 dpi, progresses to a larger number of mice (34%) at 150 dpi, indicating progressive decline of aversive memory.

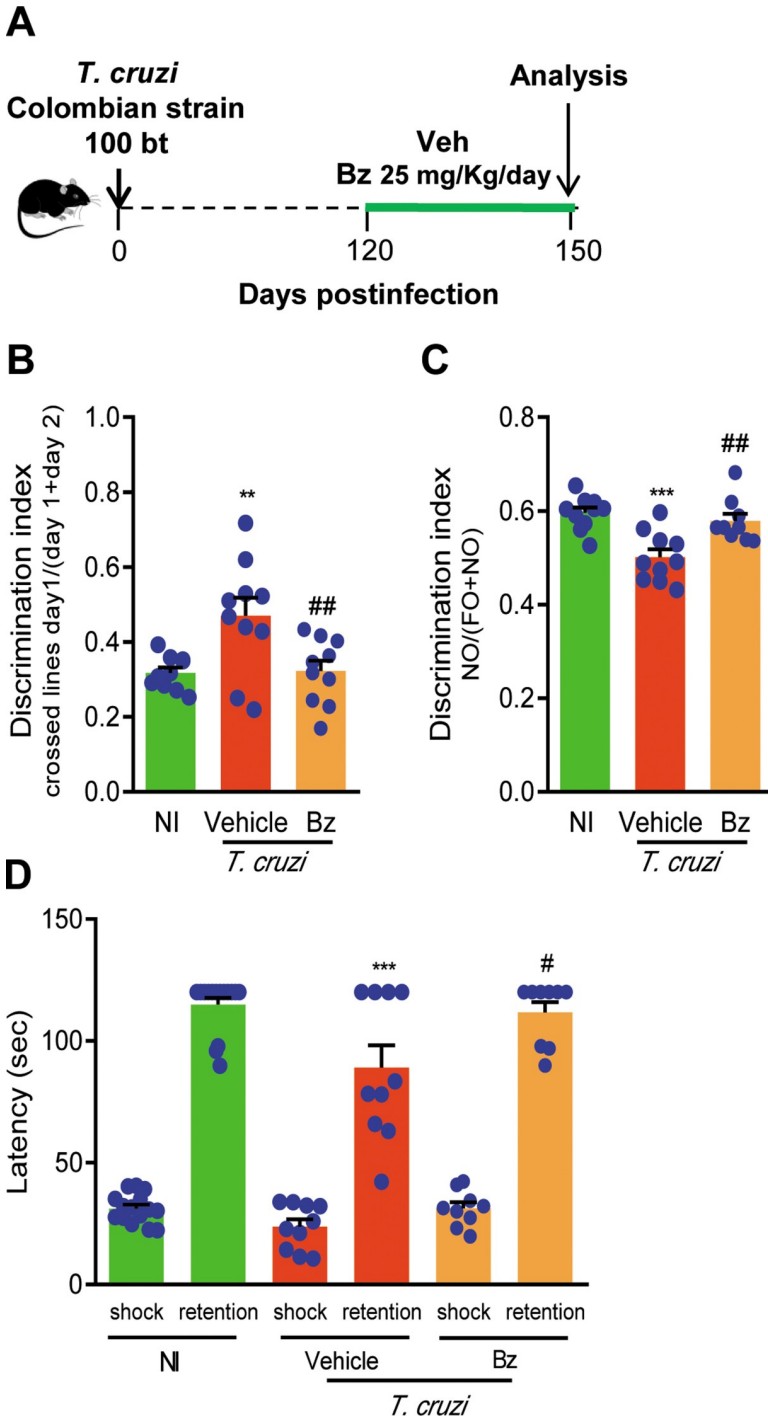

**Fig 4. Memory loss is improved by benznidazole administration to chronically *Trypanosoma cruzi*-infected C57BL/6 mice.** (**A**) Mice were infected with 100 blood trypomastigote forms of the Colombian *T. cruzi* strain and benznidazole (Bz, 0.1 mL gavage; 25 mg/Kg/day) or pyrogen-free water (Veh, 0.1 mL gavage) were administered for 30 consecutive days, from 120 to 150 dpi, when mice were tested. (**B**) The graph shows the discrimination index [number of crossed line day 2 /(number of crossed line day 1 + number of crossed line day 2)] assessed by the open field test used to evaluate habituation memory. (**C**) The graph shows the discrimination index [time exploring the novel object/ (time exploring the novel object + time exploring the familiar object)] evaluated by the novel object recognition test. (**D**) The graph shows the latency (second; sec) measured using the aversive shock evoked test to evaluate aversive memory. Each experimental group consisted of 10 NI mice and 10 *T. cruzi*-infected mice. Each circle represents an individual mouse. Data are shown as means ± SE and represent two independent experiments. Data were analyzed

using ANOVA-Bonferroni posttest. **, $p<0.01$ and ***, $p<0.001$, comparing *T. cruzi*-infected and NI mice; #, $p<0.05$ and ##, $p<0.01$, comparing Bz-treated and Veh-treated *T. cruzi*-infected.

Notably, Bz therapy initiated at 120 dpi hampered aversive memory loss at 150 dpi (Fig 4D). Therefore, Bz administration to chronically infected mice beneficially affected memory loss.

### 3.4 CNS tissue architecture is preserved in chronically *Trypanosoma cruzi*-infected C57BL/6 mice

At 150 dpi, the CNS structures were preserved in age-matched NI control mice. At 150 dpi, the CNS structures were also well-preserved and no signs of tissue damage as hemorrhage and parenchymal blood vessels inflammatory cuffs were detected, though rare inflammatory infiltrates were restricted to meninges in Veh-treated mice (Fig 5A). Importantly, the CNS tissue was devoid of neuroinflammation, as shown in tissue sections of brain areas as cortex, hippocampus, and cerebellum (Fig 5A). Further, Bz therapy did not modify the CNS tissue architecture in chronically *T. cruzi*-infected C57BL/6 mice and, moreover, abrogated

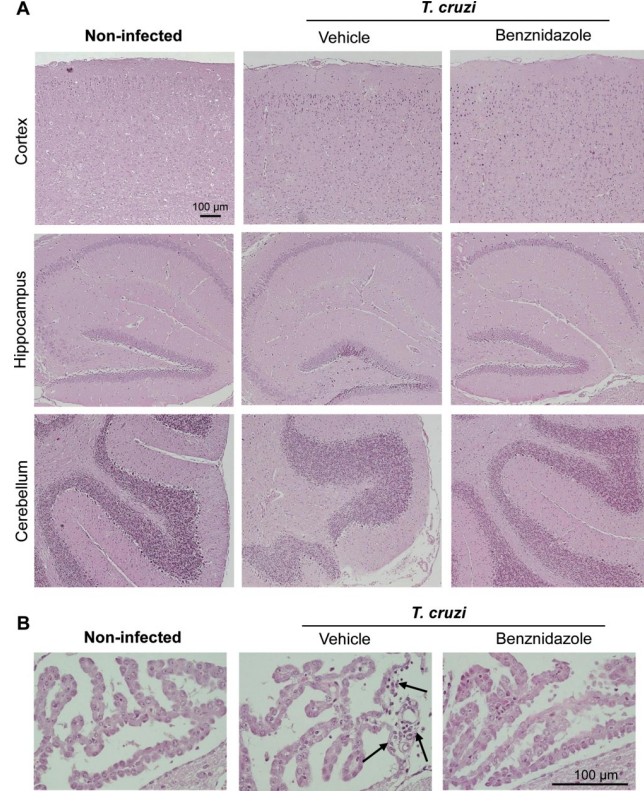

**Fig 5. CNS tissue architecture is preserved in chronically *Trypanosoma cruzi*-infected C57BL/6 mice.** Mice were infected with 100 blood trypomastigote forms of the Colombian *T. cruzi* strain and benznidazole (Bz, 0.1 mL gavage; 25 mg/Kg/day) or pyrogen-free water (Veh, 0.1 mL gavage) were administered for 30 consecutive days, from 120 to 150 dpi, when mice were euthanized, tissue collected and fixed. Tissue sections were stained with hematoxylin and eosin. (**A**) Representative sections of cerebral cortex, hippocampus, and cerebellum areas of noninfected, and Veh-treated and Bz-treated *T. cruzi*-infected mice. Bar = 100 μm. (**B**) Representative sections of choroid plexus areas of noninfected, and Veh-treated and Bz-treated *T. cruzi*-infected mice. Bar = 100 μm. Arrows show infiltration of rare mononuclear inflammatory cells in perivascular areas.

the sporadic inflammatory infiltrates in meninges (Fig 5A). At 150 dpi, Veh-treated *T. cruzi*-infected mice showed blood vessels with a few mononuclear leukocytes and rare inflammatory cells infiltrating the choroid plexus, when compared with age-matched NI controls. Again, inflammatory cells were not detected in the choroid plexus of Bz-treated infected mice (Fig 5B).

### 3.5 Therapeutic intervention with benznidazole reduces parasite load in the CNS of chronically *Trypanosoma cruzi*-infected C57BL/6 mice

Next, we challenged the influence of parasite load in the CNS in behavioral alterations in chronically infected mice. As expected, 30 days of Bz administration (from 120 to 150 dpi) to chronically Colombian-infected C57BL/6 mice efficiently reduced parasitemia (Fig 6A). Parasite DNA was detected and quantified in the hippocampus and cerebral cortex of chronically *T. cruzi*-infected C57BL/6 mice (Fig 6B and 6C). Notably, in comparison with Veh administration, Bz therapy reduced parasite load in the hippocampus and cerebral cortex of infected mice (Fig 6B and 6C).

### 3.6 Therapeutic intervention with Bz decreases oxidative stress in the cerebral cortex of chronically infected mice

To provide additional mechanistic insights on the beneficial effects of Bz therapy on memory loss, we analyzed the presence of TBARS, as a biomarker of lipid peroxidation and oxidative stress, in the CNS of chronically *T. cruzi*-infected mice. At 120 dpi (pre-therapy point) and 150 dpi (end point of therapy) CNS were collected, dissected, and analyzed. At both time points, increased TBARS levels were detected in the hippocampus and cerebral cortex of infected mice, when compared with NI controls (Fig 7A–7B). A representative experiment shows that Bz therapy reduced TBARS levels in the extracts of hippocampus in 2 out of 3 tested infected mice (Fig 7A). Importantly, Bz therapy significantly reduced TBARS levels in the extracts of cerebral cortex of *T. cruzi*-infected mice (Fig 7B), supporting that Bz therapy reduced the abnormal lipid peroxidation in the CNS of chronically infected mice.

## 4. Discussion

In the present study, we showed that chronically *T. cruzi*-infected mice present deficits of spatial habituation, novel object recognition and aversive memory recall, in the absence of neuroinflammation but with *T. cruzi* parasite persistence and increased lipid peroxidation in the CNS hippocampus and cortex. Importantly, Bz therapy interfered beneficially with cognitive impairment in a way related to reduction of parasite load systemically and in the CNS. Further, Bz-therapy decreased the levels of lipid peroxidation in the cerebral cortex.

As consequence of successful interventions targeting vector transmission, most of the CD patients in Latin America are over 45-years old [2] and may be vulnerable to aging-borne behavioral abnormalities [20, 33]. The influence of *T. cruzi* infection on behavioral alterations, particularly on memory loss, is questioned. Importantly, cognitive and memory impairments have been described in chronic CD patients, mainly detected as disturbs of comprehension and perception, orientation and attention loss, as well as impaired capacity to answer to visual-paired comparison tests [11–15]. However, it is a controversial matter as immediate and delayed memory impairment were not detected in patients with Chagas' heart disease, the main clinical form of CD [7]. Herein, we questioned whether *T. cruzi* infection may impact memory/learning process in an experimental model of chronic CD. To approach our

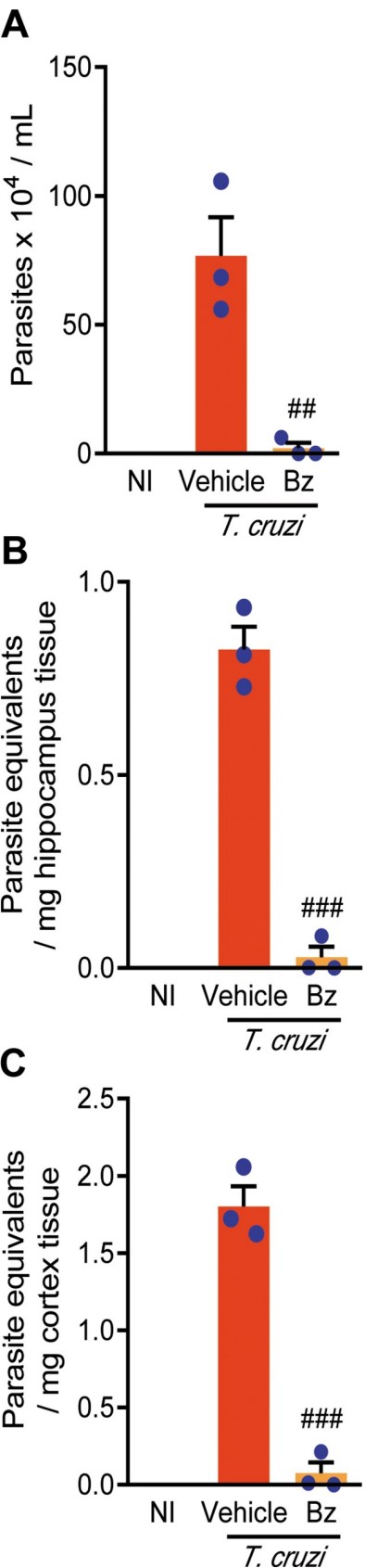

**Fig 6. Benznidazole therapy reduces parasitemia and parasite load in the central nervous system of chronically *T. cruzi*-infected C57BL/6 mice.** Mice were infected with 100 blood trypomastigote forms of the Colombian *T. cruzi* strain and benznidazole (Bz, 0.1 mL gavage; 25 mg/Kg/day) or pyrogen-free water (Veh, 0.1 mL gavage) were administered for 30 consecutive days, from 120 to 150 dpi, when parasitemia was analyzed, mice were euthanized, tissue collected, DNA extracted and qPCR performed. (**A**) Parasitemia levels. Group data for qPCR detection of *T. cruzi* satDNA in (**B**) hippocampus and (**C**) cerebral cortex. Each experimental group consisted of 3 NI mice and 3 randomly sorted *T. cruzi*-infected mice per experimental group. Each circle represents an individual mouse. Data are shown as means ± SE and represent two independent experiments. Data were analyzed using ANOVA-Bonferroni posttest. ##, $p < 0.01$ and ###, $p < 0.001$, comparing Bz-treated and Veh-treated *T. cruzi*-infected.

questions, C57BL/6 mice were infected with low inoculum of blood trypomastigotes of the Colombian type I *T. cruzi* strain, which allowed immune response to control parasite, host survival and development of the chronic phase of infection [18, 22]. Further, in comparison with age-matched non-infected controls, chronically infected mice did not display neuromuscular disorders assessed by grip strength meter test, and, therefore, suitable as an experimental model to address our issues. Non-infected C57BL/6 mice preserved the abilities to establish and evoke habituation, novel object recognition and aversive shock-evoked memory. However, in age-matched Colombian-infected mice, object recognition memory was impaired, while habituation memory was preserved, at 120 dpi. At this moment, only a low frequency (19%) of mice showed aversive memory impairment, while at 150 dpi object recognition and habituation memories were disrupted, and the frequency of mice with aversive memory impairment was increased (34%). Dissociated memory loss has been reported as a traumatic brain injury outcome, with novel object recognition memory deficit without affecting habituation and aversive memory [34]. Moreover, long-term memory evocation tests showed that in chronically *T. cruzi*-infected mice disturbs in object recognition, habituation and aversive memory tends to be progressively established, mimicking aspects of memory loss in CD

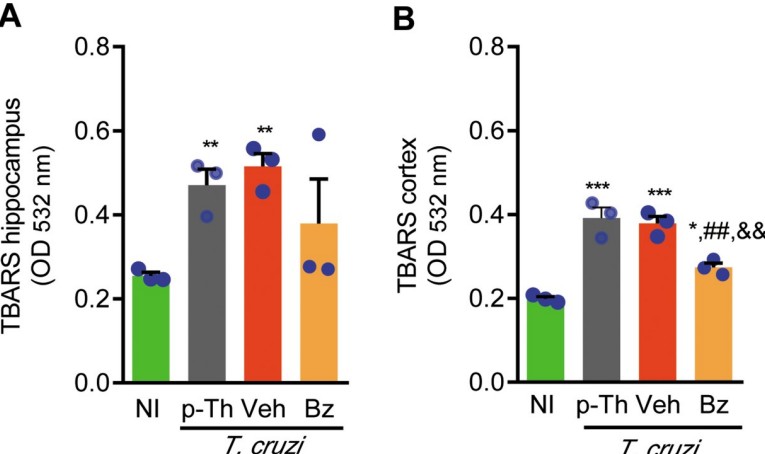

**Fig 7. Therapeutic intervention with Bz decreases oxidative stress in cerebral cortex of chronically infected mice.** Mice were infected with 100 blood trypomastigote forms of the Colombian *T. cruzi* strain and benznidazole (Bz, 0.1 mL gavage; 25 mg/Kg/day) or pyrogen-free water (Veh, 0.1 mL gavage) were administered for 30 consecutive days, from 120 to 150 dpi, when mice were euthanized, tissue collected, extracts prepared and TBARS revealed the oxidative stress marker MDA in (**B**) hippocampus and (**C**) cerebral cortex. Each experimental group consisted of 3 NI mice and 3 randomly sorted *T. cruzi*-infected mice per experimental group. Each circle represents an individual mouse. Data are shown as means ± SE and represent two independent experiments. Data were analyzed using ANOVA-Bonferroni posttest. *, $p < 0.05$, **, $p < 0.01$ and ***, $p < 0.001$, comparing *T. cruzi*-infected and NI mice; ##, $p < 0.01$, comparing Bz-treated and Veh-treated *T. cruzi*-infected. &&, $p < 0.01$, comparing Bz-treated and *T. cruzi*-infected pre-therapy (p-Th; 120 dpi).

patients [11–15]. This progressive decline of memory is a common feature of mnemonic deficits in aging and neurodegenerative disorders [20, 33]. It is important to note that these cognitive abnormalities were simultaneous with cerebral atrophy, as described in patients with chronic CD [9]. Therefore, *T. cruzi* infection may trigger and/or accelerate an aging-associated process of memory decline, here well reproduced in C57BL/6 mice chronically infected with type I parasite strain.

The association of cognitive impairment and CD is biologically plausible, as severe Chagas' heart disease frequently shows thromboembolism and congestive heart failure [2]. Conditions leading to oxygen deprivation and injury of the CNS have been associated with behavioral alterations and memory loss [35, 36]. The relation between cerebrovascular resistance evaluated by transcranial Doppler and worse cognitive scores suggests that microembolism may be responsible for a significant proportion of cognitive symptoms in chagasic and non-chagasic patients with congestive heart failure [37]. Acute *T. cruzi* infection induces cerebral microvasculopathy, with increased leukocyte-endothelium interaction (rolling and adhesion) and microvascular platelet-leukocyte aggregates [38, 39]. However, in chronically Colombian-infected mice, the CNS blood vessels are not activated, being refractory to cell adhesion [39]. Nevertheless, the contribution of heart disease to memory impairment cannot be ruled out as the Colombian-infected C57BL/6 mice present chronic chagasic cardiomyopathy with electrical abnormalities and reduced ventricular function [22, 40], thus limiting the interpretation of our findings. However, in a group of chronic CD patients cognitive impairment was associated with *T. cruzi* infection, but not mediated by CD-related electrocardiographic abnormalities [10], supporting that these could be dissociated processes.

Administration of the trypanocide drug Bz to chronically infected CD patients reduces parasite load in blood and does not aggravate the disease. However, Bz therapy presents controversial results, with beneficial or no measurable effects, in Chagas' heart disease outcome [41–45]. Bz therapy initiated after parasitemia control, in the early chronic phase of *T. cruzi* infection, reduced the severity of chronic heart disease at 8 months of infection [46]. When Bz was administered to chronically infected mice to reverse heart disease, the beneficial effects were restricted to improve average heart rate and reduce the frequency of mice with arrhythmias and second-degree atrioventricular blockage [24]. Importantly, administration of the full recommended dose of Bz (100 mg/Kg/day) in the acute phase of experimental Colombian infection prevented depressive-like behavior in the chronic phase of infection [17]. Here, we showed that memory deficits described in chronically infected mice were beneficially impacted by Bz therapy. The administration of reduced Bz dosage, one fourth of the recommended dose [24], initiated at 120 dpi for short period (30 consecutive days) prevented the habituation memory impairment. Moreover, Bz therapy reversed object recognition memory loss and hampered progression of the aversive memory impairment. Therefore, it is the first demonstration that Bz administration ameliorated and, even more, reversed cognitive abnormalities.

In CD patients, the cognitive impairment lacks histopathological evidence and occurs in the absence of neuroinflammation [8, 9]. Nonexistence of neuroinflammation is also a hallmark of experimental chronic *T. cruzi* infection [39, 47]. Here, we bring evidence that in Colombian-infected C57BL/6 mice, memory impairment occurs in the absence of inflammation in the CNS cortex, hippocampus, and cerebellum, crucial areas for memory functions [33, 48]. In the adopted experimental model of chronic infection of C57BL/6 mice, rare inflammatory cells were restricted to meninges and choroid plexus, corroborating studies using the Colombian *T. cruzi* strain for the acute infection of Swiss mice and the chronic infection of C3H/He mice [39, 47, 49, 50]. Although in low numbers and restricted to meninges and choroid plexus, one cannot rule out the participation of these infiltrating mononuclear cells and

their products, as cytokines and chemokines, in the cognitive impairment detected in chronically *T. cruzi*-infected mice.

The *T. cruzi* parasite DNA was detected in the CNS hippocampus and cortex areas of chronically Colombian-infected mice. Previously, low-grade parasitism was revealed as *T. cruzi*-antigen positive cells randomly scattered throughout the CNS parenchyma of acute and chronically infected mice [39, 47, 50]. Further, in HIV-infected immunosuppressed chronic CD patients the CNS is the main site of parasitism reactivation [51]. Thus, these data support parasite persistence in the CNS tissue in chronic *T. cruzi* infection in humans and experimental models, being able to contribute to behavioral alterations, as memory dysfunction. In another neuroinfection, caused by the intracellular parasite *Toxoplasma gondii*, behavioral changes as psychomotor alterations and decreased memory performance have been described and related with parasite persistence in the CNS [52]. Therefore, we asked the contribution of parasite persistence in the CNS to memory impairment in the chronic phase of experimental infection, using Bz as trypanocide drug. Previously, we have shown that Bz therapy in the acute *T. cruzi* infection prevented depressive-like behavior in early chronic infection [17], supporting the idea that *T. cruzi* infection may trigger behavioral alterations. Here, our data support that administration of a low dose of Bz for a short period to chronically *T. cruzi*-infected mice showed beneficial effects on memory loss, as prevented habituation memory impairment, reversed the object recognition memory loss, and hampered progression of aversive memory disruption. These beneficial effects were associated with decreased parasitemia, supporting the systemically action of Bz, and reduced parasite load in the hippocampus and cerebral cortex areas, showing the effectiveness of Bz in the CNS parasite control. In the BENEFIT study, in which chronic cardiac CD patients were treated with Bz or placebo, it was shown that the reduction in blood parasite load did not significantly affect the clinical deterioration of the patients [43]. Bz administration to chronically Colombian-infected C57BL/6 mice reduced parasitemia and heart parasite load but showed limited beneficial effects on cardiac disease [24]. Interestingly, in a model of septic shock Bz showed down-regulatory effect on tumor necrosis factor (TNF) production selectively linked to the nuclear factor NF-kB and mitogen-activated protein kinase (MAPK) pathways [53]. Further, Bz administration to chronically *T. cruzi*-infected mice also reduced TNF expression in the heart tissue [24]. Astrocyte-born TNF may contribute to *T. cruzi* persistence mainly in astrocytes, as parasite-positive spots in the CNS [47, 54]. However, in the CNS TNF expression may be restricted to these parasite-positive areas, where TNF may be rapidly consumed. Since TNF upregulation was undetectable in the CNS of the Colombian-infected mice [17], the effects of Bz on TNF expression in the brain tissue could not be explored. Therefore, besides the direct effect on intracellular parasite forms, the beneficial effects of Bz in reducing the CNS parasite load may reside in disrupting the autocrine cytokine production and, therefore, TNF signaling. Notably, the administration of anti-TNF and pentoxiphyline, a modulator of TNF receptor 1 (TNFR1) expression, to chronically *T. cruzi*-infected mice supports that signaling via TNF/TNFR1 may take part in depressive-like behavior [17]. Thus, a point to be further explored is the contribution of TNF/TNFR signaling in *T. cruzi* infection-associated memory loss. Crucially, there is an emerging comprehension of how dysregulation of cytokine networks is associated with neurodegenerative diseases and memory impairment in humans and animal models [55].

Multiple or sequential processes may contribute to memory deficits in chronically *T. cruzi*-infected mice. In Alzheimer disease, behavioral changes have been associated with oxidative stress, characterized as augmented lipid peroxidation in the cerebral cortex [19, 32]. Oxidative damage induces a cascade of downstream reactive oxygen species, some transient and other

downstream substances that accumulate in the CNS as malondialdehyde, a TBARS that may be detected in tissue extracts [56]. In mice acutely infected with type II *T. cruzi* strain, reduced response of the CNS endothelial cells to acetylcholine associated to enhanced detection of TBARS, was restricted to the parasitemia peak [38]. Further, in acutely *T. cruzi*-infected Swiss mice, aversive memory impairment has been shown in association with oxidative stress, revealed as increased cerebral acetylcholinesterase activity [57]. Here, we showed in chronically infected mice a relation of memory loss and enhanced oxidative stress, revealed as increased TBARS in the CNS hippocampus and cortex areas pre-therapy (120 dpi) and in vehicle-treated (150 dpi) mice. In the CNS, this increase in oxidative stress may result of the action of the parasite directly leading to cell death or indirectly inducing production of inflammatory mediators into this tissue. However, one could not exclude the leakage of reactive substances into the CNS parenchyma, since in chronically *T. cruzi*-infected rats increased TBARS in plasma may unveil a systemic oxidative stress [58]. Administration of Bz (40 mg/Kg/day) elicited oxidative stress and decreased antioxidant machinery in rat hepatocytes, which may contribute to Bz toxicity [59]. However, it does not seem to be the case in our study, since the levels of TBARS were not increased in the hippocampus and showed a significant reduction in the cerebral cortex after 30 consecutive days of Bz therapy, when memory deficits were beneficially impacted. These conflicting data may be explained as Bz is metabolized in liver, the brain is less permeable to orally administered Bz, we used a model of chronic *T. cruzi* infection in mouse and administered a low dose of Bz, which may reduce the toxic effects of this trypanocidal drug, in accordance with our initial goal [24, 60].

In conclusion, our results support that the nervous form of chronic CD occurs in the absence of neuroinflammation but associated with parasite persistence and increased oxidative stress in the CNS, showing as clinical outcome behavioral changes as long-term memory impairment. Although with some limitations, mostly related to the molecular levels of Bz action in the CNS, our present findings open a new pathway to be explored using Bz in the chronic phase of CD not only aiming to treat the cardiac form, but also to ameliorate cognitive disorders and improve the quality of life of CD patients.

## Supporting information

**S1 Checklist. The arrive essential 10: Author checklist.**
(PDF)

**S1 Fig. Beneficial effect of benznidazole administration on splenomegaly of chronically *Trypanosoma cruzi*-infected C57BL/6 mice.** C57BL/6 mice were infected with 100 blood trypomastigotes of the Colombian strain of *T. cruzi*, and treated with Veh or Bz, as described in legend of Fig 3. (**A**) Body weight (g). (**B**) Relative spleen weight (mg/g). Data represent two independent experiments with 3 NI controls and 5 infected mice per group. Each circle represents an individual mouse. Data are shown as means ± SE and were analyzed using ANOVA--Bonferroni posttest. *, $p<0.05$ and ***, $p<0.001$, comparing *T. cruzi*-infected and NI mice; ##, $p<0.01$, comparing Bz-treated and Veh-treated *T. cruzi*-infected.
(TIF)

## Author Contributions

**Conceptualization:** Glaucia Vilar-Pereira, Leda Castaño Barrios, Andrea Alice da Silva, Joseli Lannes-Vieira.

**Data curation:** Andrea Alice da Silva, Otacílio Cruz Moreira, Joseli Lannes-Vieira.

**Formal analysis:** Glaucia Vilar-Pereira, Leda Castaño Barrios, Andrea Alice da Silva, Angelica Martins Batista, Isabela Resende Pereira, Otacílio Cruz Moreira, Constança Britto, Hílton Antônio Mata dos Santos, Joseli Lannes-Vieira.

**Funding acquisition:** Joseli Lannes-Vieira.

**Investigation:** Angelica Martins Batista, Isabela Resende Pereira.

**Methodology:** Glaucia Vilar-Pereira, Leda Castaño Barrios, Andrea Alice da Silva, Otacílio Cruz Moreira, Constança Britto, Hílton Antônio Mata dos Santos.

**Supervision:** Joseli Lannes-Vieira.

**Validation:** Leda Castaño Barrios.

**Writing – original draft:** Glaucia Vilar-Pereira, Leda Castaño Barrios.

**Writing – review & editing:** Joseli Lannes-Vieira.

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
