## [Decision Letter · Decision Letter 0]

20 Nov 2020

PONE-D-20-30457

Memory impairment in chronic experimental Chagas disease: benznidazole therapy reversed cognitive deficit in association with reduction of parasite load and oxidative stress in the nervous tissue

PLOS ONE

Dear Dr. Lannes-Vieira,

Thank you for submitting your manuscript to PLOS ONE. After careful consideration, we feel that it has merit but does not fully meet PLOS ONE’s publication criteria as it currently stands. Therefore, we invite you to submit a revised version of the manuscript that addresses the points raised during the review process.

I have now received reports from two expert referees on your manuscript "Memory impairment in chronic experimental Chagas disease: benznidazole therapy reversed cognitive deficit in association with reduction of parasite load and oxidative stress in the nervous tissue". Both reviewers found your work of interest; however, they raised some minor issues that will need to be addressed before further consideration. Mostly, details in the methods and results sections are lacking, and there are some important inconsistencies between reports of findings and explanation of methods. Altogether, the manuscript should be revised for grammar and typos mistakes.

I am enclosing the comments of the referees for a complete description of issues. 

We look forward to receiving your revised manuscript.

Kind regards,

Laurence Coutellier, PhD

Academic Editor

PLOS ONE

Reviewers' comments:

Reviewer's Responses to Questions

**Comments to the Author**

1. Is the manuscript technically sound, and do the data support the conclusions?

Reviewer #1: Yes

Reviewer #2: Yes

2. Has the statistical analysis been performed appropriately and rigorously? 

Reviewer #1: Yes

Reviewer #2: N/A

3. Have the authors made all data underlying the findings in their manuscript fully available?

Reviewer #1: Yes

Reviewer #2: Yes

4. Is the manuscript presented in an intelligible fashion and written in standard English?

Reviewer #1: Yes

Reviewer #2: Yes

5. Review Comments to the Author

Reviewer #1: Authors successfully show that T. cruzi infection leads to impairment of memory that is rescued by Bz therapy compared to non-infected controls. They could however benefit from revising the manuscript to make sure that terms and abbreviations are defined when 1st introduced. A few grammar changes could also be beneficial

Reviewer #2: The authors evaluated the effect of the experimental chronic T. cruzi infection (using a low inoculum of blood trypomastigotes of the Colombian strain) on the memory/learning process of C57BL/ 6 mice submitted or not for BZ treatment. They observed that BZ treatment hampered the progression of the habituation and aversive memory loss and reversed the recognition memory impairment caused by T. cruzi infection. These findings also were followed by reduction of parasitemia, parasite load and oxidative stress in the CNS.

The manuscript in general is well-written. The hypothesis and main research question are clearly presented. Experiments design and results are described in sufficient detail and the discussion are rich and very well elaborated. The results content supports the main conclusion. I will list only minor issues.

1- In methods section (Lines 126-128). Organize better the explanation about the group composition. It is very unclear for the reader.

2- Line 139. The serial passages were performed in which mice?

3- Regarding the qPCR. Did the authors use a murine normalizer primer? (i.e., b-act) How did the authors normalize the data gotten with Cruzi primers?

4- I got confused about the mice ages used in the experiments. In methods section, the authors clearly say they have used female of 5-7-weeks old. When you go to results section, line 360, the authors say “CNS structures were preserved in elderly NI control” but there is no description of the use of elderly mice in the methods nor in the legends. Further up, in the line 418 (discussion section) they mention: “Non-infected elderly C57BL/6 mice preserved the abilities to establish and evoke habituation, novel object recognition and aversive shock-evoked memory” again, these findings were not mentioned in the results and no description of elderly mice in methods nor in the legends. I also thought that these findings could be from the literature. In that case, they must be followed by reference.

5- Results section (Line 306). Rewrite please, very confuse. Also, in line 308 it seems that the authors did not finish the sentence: “…female mice were infected with 100 trypomastigote forms of the Colombian strain of T. cruzi and weekly analyzed (Fig 1).” Analyzed for what?

6- Lines 342-344. The information is very unclear. BZ therapy did not impact body weight of which group? It is very confusing for the reader.

7- I missed a better explanation in the results section why the authors evaluated CNS parasite load (lines 375-380). This information is clearer in the discussion section, for example, the authors shortly mentioned they are evaluating the contribution of parasite persistence in CNS to memory impairment. A sentence like this should come in the first paragraph before the authors start to describe the findings in the line 375 to create a link between the previous results and parasite load evaluation.

8- Line 386: Please describe better the experiment. What is difference between 120dpi (pre-therapy) and 150 dpi BZ treatments? It is unclear.

9- Discussion section. Lines 457-458. I do not understand what the authors mean here. Please clarify.

10- Line 511. Fix the punctuation after the word ‘further’.

6. PLOS authors have the option to publish the peer review history of their article (what does this mean?). If published, this will include your full peer review and any attached files.

Reviewer #1: No

Reviewer #2: No

---

## [Author Response · Author response to Decision Letter 0]

4 Dec 2020

Answers to reviewers 

Manuscript Number: PONE-D-20-30457

Manuscript Title: Memory impairment in chronic experimental Chagas disease: benznidazole therapy reversed cognitive deficit in association with reduction of parasite load and oxidative stress in the nervous tissue

Dear Editor,

Firstly, we would like to thank the reviewers’ comments 

Please find below point-by point answers to reviewers ‘comments.

Best regards,

Joseli Lannes

Reviewer #1:

Minor revisions

1. Line 52 please define Veh-treated infected mice – modified – line 50.

2. Lines 132-135 please define the abbreviations – clarified – line 133.

3. Line 202 please define NI – this sentence was modified – line 204-205 – NI has been previously defined - line 164.

4. Line 360 tell the reader here how you define ‘elderly’ – this was corrected. – line 365.

Thank you for bringing this issue. It was a misinterpretation, and your comment gave us the opportunity to clarify this point. In fact, the animals analysed in the present study were not elderly mice, but “mature adults”. Therefore, the term “elderly” has been withdrawn. We received the animal when they were 5-7-week old, kept for adaptation for 10-14 days. They were infected when 7-9-week-old (2 months - young adults) and analyzed at 120- or 150-days post-infection, (6-7-month-old mice). Thus, our mice were “mature adults”, as described by Flurkey K, Currer JM, Harrison DE. 2007. The Mouse in Aging Research. In The Mouse in Biomedical Research 2nd Edition. Fox JG, et al, editors. American College Laboratory Animal Medicine (Elsevier), Burlington, MA. pp. 637–672. 

5. Line 364 kindly check the grammar for ‘tissues section’ sentence – corrected – lines 369-370.

6. Line 416 remind the reader how you determined ‘chronically infected mice did not display neuromuscular disorders’ – the sentence was modified – “…chronically infected mice did not display neuromuscular disorders assessed by grip strength meter test …” – lines 424-425. 

7. Line 509 define ‘NF-kB- and MAPK’ – Rewritten “Interestingly, in a model of septic shock Bz showed down-regulatory effect on tumor necrosis factor (TNF) production selectively linked to the nuclear factor NF-kB and mitogen-activated protein kinase (MAPK) pathways” – lines 518-520. 

8. Line 520 define ‘TNFR1’ – defined – line 530-531

9. Line 532 define a ‘TBARS’ – It has been defined at line 286 and used several times in the Result and Discussion sections. 

Reviewer #2:

1- In methods section (Lines 126-128). Organize better the explanation about the group composition. It is very unclear for the reader. – the description of the data was better organized and improved – lines 123 – 130. 

2- Line 139. The serial passages were performed in which mice? – it was rephrased – “…and kept by serial passages in C57BL/6 female mice…” – line 141.

3- Regarding the qPCR. Did the authors use a murine normalizer primer? (i.e., b-act) How did the authors normalize the data gotten with Cruzi primers?

Yes. As explained at the Materials and methods section (subitem 2.10 – line 247-282), a commercial TaqMan system targeting mouse GAPDH (Glyceraldehyde-3-phosphate dehydrogenase VIC/TAMRA-labelled, Mm99999915g1, Thermo Fisher Scientific, USA) was used to quantify the brain tissue mg equivalents. The cortex or hippocampus mass equivalents were estimated by absolute quantification, using a standard curve generated by the serial dilution of DNA extracted from mouse cortex or hippocampus spiked with T. cruzi. Thus, the parasite load was normalized by the tissue mg equivalents, by dividing the T. cruzi quantity by the mass of mice tissue equivalents (cortex or hippocampus).

In order to clarify the normalization of parasite load, the following sentence was added at lines 279-280, at the new version of the manuscript: “…, by dividing the T. cruzi quantity by the mass of mice tissue equivalents (cortex or hippocampus)”.

4- I got confused about the mice ages used in the experiments. In methods section, the authors clearly say they have used female of 5-7-weeks old. When you go to results section, line 360, the authors say “CNS structures were preserved in elderly NI control” but there is no description of the use of elderly mice in the methods nor in the legends. Further up, in the line 418 (discussion section) they mention: “Non-infected elderly C57BL/6 mice preserved the abilities to establish and evoke habituation, novel object recognition and aversive shock-evoked memory” again, these findings were not mentioned in the results and no description of elderly mice in methods nor in the legends. I also thought that these findings could be from the literature. In that case, they must be followed by reference.

Thank you for bringing this issue. It was a misinterpretation, and your comment gave us the opportunity to clarify this point. In fact, the animals analysed in the present study were not elderly mice, but “mature adults”. Therefore, the term “elderly” has been withdrawn. We received the animal when they were 5-7-week old, kept for adaptation 10-14 days. They were infected when 7-9-week-old (2 months - young adults) and analyzed at 120- or 150-days post-infection, (6-7-month-old mice). Thus, our mice were “mature adults”, as described by Flurkey K, Currer JM, Harrison DE. 2007. The Mouse in Aging Research. In The Mouse in Biomedical Research 2nd Edition. Fox JG, et al, editors. American College Laboratory Animal Medicine (Elsevier), Burlington, MA. pp. 637–672. 

There is no available data on elderly T. cruzi-infected mice in the literature. We have planned to approach this idea in the next study, analysing mice after 270 days post-infection (11 month-old mice; middle-aged animals), when most infected mice are still alive (70-80%). However, this study has been postponed by the epidemiological scenario of Covid-19. 

5- Results section (Line 306). Rewrite please, very confuse. Also, in line 308 it seems that the authors did not finish the sentence: “…female mice were infected with 100 trypomastigote forms of the Colombian strain of T. cruzi and weekly analyzed (Fig 1).” Analyzed for what?

The sentence was rewritten. Lines 310-312 – “Out of 92 C57BL/6 female mice, 25 were non-infected (NI) controls and 67 were infected with 100 trypomastigote forms of the Colombian strain of T. cruzi and weekly monitored for parasitemia, body weight and clinical follow-up (Fig 1).”

6- Lines 342-344. The information is very unclear. BZ therapy did not impact body weight of which group? It is very confusing for the reader.

This sentence was rephrased. Lines 347-349 – “At 150 dpi, Bz therapy did not impact body weight (S2A Fig.), when compared to NI controls and Veh-treated infected and likened to T. cruzi-infected mice pre-therapy (at 120 dpi).”

7- I missed a better explanation in the results section why the authors evaluated CNS parasite load (lines 375-380). This information is clearer in the discussion section, for example, the authors shortly mentioned they are evaluating the contribution of parasite persistence in CNS to memory impairment. A sentence like this should come in the first paragraph before the authors start to describe the findings in the line 375 to create a link between the previous results and parasite load evaluation.

Lines 380-381 – the sentence added – “Next, we challenged the influence of parasite load in the CNS in behavioral alterations in chronically infected mice”.

8- Line 386: Please describe better the experiment. What is difference between 120dpi (pre-therapy) and 150 dpi BZ treatments? It is unclear.

These time points of analysis were better described in the Materials and methods section (lines 123-130) and in the Results section (lines 392-394).

9- Discussion section. Lines 457-458. I do not understand what the authors mean here. Please clarify.

We rephrased it – Lines 465-467 – “Bz therapy initiated after parasitemia control, in the early chronic phase of T. cruzi infection, reduced the severity of chronic heart disease at 8 months of infection [46]”.

10- Line 511. Fix the punctuation after the word ‘further’. – corrected. Line 521.

Answers to reviewers 

Manuscript Number: PONE-D-20-30457

Manuscript Title: Memory impairment in chronic experimental Chagas disease: benznidazole therapy reversed cognitive deficit in association with reduction of parasite load and oxidative stress in the nervous tissue

Dear Editor,

Firstly, we would like to thank the reviewers’ comments 

Please find below point-by point answers to reviewers ‘comments.

Best regards,

Joseli Lannes

Reviewer #1:

Minor revisions

1. Line 52 please define Veh-treated infected mice – modified – line 50.

2. Lines 132-135 please define the abbreviations – clarified – line 133.

3. Line 202 please define NI – this sentence was modified – line 204-205 – NI has been previously defined - line 164.

4. Line 360 tell the reader here how you define ‘elderly’ – this was corrected. – line 365.

Thank you for bringing this issue. It was a misinterpretation, and your comment gave us the opportunity to clarify this point. In fact, the animals analysed in the present study were not elderly mice, but “mature adults”. Therefore, the term “elderly” has been withdrawn. We received the animal when they were 5-7-week old, kept for adaptation for 10-14 days. They were infected when 7-9-week-old (2 months - young adults) and analyzed at 120- or 150-days post-infection, (6-7-month-old mice). Thus, our mice were “mature adults”, as described by Flurkey K, Currer JM, Harrison DE. 2007. The Mouse in Aging Research. In The Mouse in Biomedical Research 2nd Edition. Fox JG, et al, editors. American College Laboratory Animal Medicine (Elsevier), Burlington, MA. pp. 637–672. 

5. Line 364 kindly check the grammar for ‘tissues section’ sentence – corrected – lines 369-370.

6. Line 416 remind the reader how you determined ‘chronically infected mice did not display neuromuscular disorders’ – the sentence was modified – “…chronically infected mice did not display neuromuscular disorders assessed by grip strength meter test …” – lines 424-425. 

7. Line 509 define ‘NF-kB- and MAPK’ – Rewritten “Interestingly, in a model of septic shock Bz showed down-regulatory effect on tumor necrosis factor (TNF) production selectively linked to the nuclear factor NF-kB and mitogen-activated protein kinase (MAPK) pathways” – lines 518-520. 

8. Line 520 define ‘TNFR1’ – defined – line 530-531

9. Line 532 define a ‘TBARS’ – It has been defined at line 286 and used several times in the Result and Discussion sections. 

Reviewer #2:

1- In methods section (Lines 126-128). Organize better the explanation about the group composition. It is very unclear for the reader. – the description of the data was better organized and improved – lines 123 – 130. 

2- Line 139. The serial passages were performed in which mice? – it was rephrased – “…and kept by serial passages in C57BL/6 female mice…” – line 141.

3- Regarding the qPCR. Did the authors use a murine normalizer primer? (i.e., b-act) How did the authors normalize the data gotten with Cruzi primers?

Yes. As explained at the Materials and methods section (subitem 2.10 – line 247-282), a commercial TaqMan system targeting mouse GAPDH (Glyceraldehyde-3-phosphate dehydrogenase VIC/TAMRA-labelled, Mm99999915g1, Thermo Fisher Scientific, USA) was used to quantify the brain tissue mg equivalents. The cortex or hippocampus mass equivalents were estimated by absolute quantification, using a standard curve generated by the serial dilution of DNA extracted from mouse cortex or hippocampus spiked with T. cruzi. Thus, the parasite load was normalized by the tissue mg equivalents, by dividing the T. cruzi quantity by the mass of mice tissue equivalents (cortex or hippocampus).

In order to clarify the normalization of parasite load, the following sentence was added at lines 279-280, at the new version of the manuscript: “…, by dividing the T. cruzi quantity by the mass of mice tissue equivalents (cortex or hippocampus)”.

4- I got confused about the mice ages used in the experiments. In methods section, the authors clearly say they have used female of 5-7-weeks old. When you go to results section, line 360, the authors say “CNS structures were preserved in elderly NI control” but there is no description of the use of elderly mice in the methods nor in the legends. Further up, in the line 418 (discussion section) they mention: “Non-infected elderly C57BL/6 mice preserved the abilities to establish and evoke habituation, novel object recognition and aversive shock-evoked memory” again, these findings were not mentioned in the results and no description of elderly mice in methods nor in the legends. I also thought that these findings could be from the literature. In that case, they must be followed by reference.

Thank you for bringing this issue. It was a misinterpretation, and your comment gave us the opportunity to clarify this point. In fact, the animals analysed in the present study were not elderly mice, but “mature adults”. Therefore, the term “elderly” has been withdrawn. We received the animal when they were 5-7-week old, kept for adaptation 10-14 days. They were infected when 7-9-week-old (2 months - young adults) and analyzed at 120- or 150-days post-infection, (6-7-month-old mice). Thus, our mice were “mature adults”, as described by Flurkey K, Currer JM, Harrison DE. 2007. The Mouse in Aging Research. In The Mouse in Biomedical Research 2nd Edition. Fox JG, et al, editors. American College Laboratory Animal Medicine (Elsevier), Burlington, MA. pp. 637–672. 

There is no available data on elderly T. cruzi-infected mice in the literature. We have planned to approach this idea in the next study, analysing mice after 270 days post-infection (11 month-old mice; middle-aged animals), when most infected mice are still alive (70-80%). However, this study has been postponed by the epidemiological scenario of Covid-19. 

5- Results section (Line 306). Rewrite please, very confuse. Also, in line 308 it seems that the authors did not finish the sentence: “…female mice were infected with 100 trypomastigote forms of the Colombian strain of T. cruzi and weekly analyzed (Fig 1).” Analyzed for what?

The sentence was rewritten. Lines 310-312 – “Out of 92 C57BL/6 female mice, 25 were non-infected (NI) controls and 67 were infected with 100 trypomastigote forms of the Colombian strain of T. cruzi and weekly monitored for parasitemia, body weight and clinical follow-up (Fig 1).”

6- Lines 342-344. The information is very unclear. BZ therapy did not impact body weight of which group? It is very confusing for the reader.

This sentence was rephrased. Lines 347-349 – “At 150 dpi, Bz therapy did not impact body weight (S2A Fig.), when compared to NI controls and Veh-treated infected and likened to T. cruzi-infected mice pre-therapy (at 120 dpi).”

7- I missed a better explanation in the results section why the authors evaluated CNS parasite load (lines 375-380). This information is clearer in the discussion section, for example, the authors shortly mentioned they are evaluating the contribution of parasite persistence in CNS to memory impairment. A sentence like this should come in the first paragraph before the authors start to describe the findings in the line 375 to create a link between the previous results and parasite load evaluation.

Lines 380-381 – the sentence added – “Next, we challenged the influence of parasite load in the CNS in behavioral alterations in chronically infected mice”.

8- Line 386: Please describe better the experiment. What is difference between 120dpi (pre-therapy) and 150 dpi BZ treatments? It is unclear.

These time points of analysis were better described in the Materials and methods section (lines 123-130) and in the Results section (lines 392-394).

9- Discussion section. Lines 457-458. I do not understand what the authors mean here. Please clarify.

We rephrased it – Lines 465-467 – “Bz therapy initiated after parasitemia control, in the early chronic phase of T. cruzi infection, reduced the severity of chronic heart disease at 8 months of infection [46]”.

10- Line 511. Fix the punctuation after the word ‘further’. – corrected. Line 521.

---

## [Decision Letter · Decision Letter 1]

16 Dec 2020

Memory impairment in chronic experimental Chagas disease: benznidazole therapy reversed cognitive deficit in association with reduction of parasite load and oxidative stress in the nervous tissue

PONE-D-20-30457R1

Dear Dr. Lannes,

We’re pleased to inform you that your manuscript has been judged scientifically suitable for publication and will be formally accepted for publication once it meets all outstanding technical requirements.

Kind regards,

Laurence Coutellier, PhD

Academic Editor

PLOS ONE

Reviewers' comments:

Reviewer's Responses to Questions

**Comments to the Author**

1. If the authors have adequately addressed your comments raised in a previous round of review and you feel that this manuscript is now acceptable for publication, you may indicate that here to bypass the “Comments to the Author” section, enter your conflict of interest statement in the “Confidential to Editor” section, and submit your "Accept" recommendation.

Reviewer #1: All comments have been addressed

Reviewer #2: All comments have been addressed

2. Is the manuscript technically sound, and do the data support the conclusions?

Reviewer #1: Yes

Reviewer #2: Yes

3. Has the statistical analysis been performed appropriately and rigorously? 

Reviewer #1: Yes

Reviewer #2: Yes

4. Have the authors made all data underlying the findings in their manuscript fully available?

Reviewer #1: Yes

Reviewer #2: Yes

5. Is the manuscript presented in an intelligible fashion and written in standard English?

Reviewer #1: Yes

Reviewer #2: Yes

6. Review Comments to the Author

Reviewer #1: All suggestions have been attended to satisfactorily. May I ask the authors to make sure they have used the same font color throughout the manuscript, and to please attend to line 50 and write vehicle-treated (veh-treated) so the reader knows to expect veh-treated later

Reviewer #2: (No Response)

7. PLOS authors have the option to publish the peer review history of their article (what does this mean?). If published, this will include your full peer review and any attached files.

Reviewer #1: No

Reviewer #2: **Yes: **Sandra C Rocha

---

## [Editor Report · Acceptance letter]

23 Dec 2020

PONE-D-20-30457R1 

Memory impairment in chronic experimental Chagas disease: benznidazole therapy reversed cognitive deficit in association with reduction of parasite load and oxidative stress in the nervous tissue 

Dear Dr. Lannes-Vieira:

I'm pleased to inform you that your manuscript has been deemed suitable for publication in PLOS ONE. Congratulations! Your manuscript is now with our production department. 

Kind regards, 

on behalf of

Dr. Laurence Coutellier 

Academic Editor

PLOS ONE